# A Comprehensive Framework to Evaluate the Effects of Anterior Cruciate Ligament Injury and Reconstruction on Graft and Cartilage Status through the Analysis of MRI T2 Relaxation Time and Knee Laxity: A Pilot Study

**DOI:** 10.3390/life11121383

**Published:** 2021-12-10

**Authors:** Gregorio Marchiori, Giorgio Cassiolas, Matteo Berni, Alberto Grassi, Giacomo Dal Fabbro, Milena Fini, Giuseppe Filardo, Stefano Zaffagnini, Nicola Francesco Lopomo

**Affiliations:** 1Complex Structure of Surgical Sciences and Technologies, IRCCS Istituto Ortopedico Rizzoli, 40136 Bologna, Italy; gregorio.marchiori@ior.it (G.M.); milena.fini@ior.it (M.F.); 2Department of Information Engineering, University of Brescia, 25121 Brescia, Italy; g.cassiolas@unibs.it (G.C.); nicola.lopomo@unibs.it (N.F.L.); 3Medical Technology Laboratory, IRCCS Istituto Ortopedico Rizzoli, 40136 Bologna, Italy; 42nd Orthopedic and Traumatologic Clinic, IRCCS Istituto Ortopedico Rizzoli, 40136 Bologna, Italy; alberto.grassi@ior.it (A.G.); giacomo.dalfabbro@ior.it (G.D.F.); stefano.zaffagnini@ior.it (S.Z.); 5Applied and Translational Research Center, IRCCS Istituto Ortopedico Rizzoli, 40136 Bologna, Italy; giuseppe.filardo@ior.it

**Keywords:** anterior cruciate ligament tear, anterior cruciate ligament reconstruction, quantitative magnetic resonance imaging, T2 mapping, graft maturation, knee cartilage, knee functionality, knee laxity, healing process

## Abstract

Background: Anterior cruciate ligament (ACL) tear represents a common orthopedic traumatic issue that often leads to an early development of osteoarthritis. To improve the diagnostic and prognostic techniques involved in the assessment of the joint after the trauma and during the healing process, the present work proposes a multi-parametric approach that aims to investigate the relationship between joint function and soft tissue status before and after ACL reconstruction. Methods: Thirteen consecutive patients who underwent ACL reconstruction were preliminarily enrolled in this study. Joint laxity assessment as well as magnetic resonance imaging with T2 mapping were performed in the pre-operative stage, at four and 18 months after surgery to acquire objective information to correlate knee function and soft tissue condition. Results: Correlations were found between graft and cartilage T2 signal, suggesting an interplay between these tissues within the knee joint. Moreover, graft maturation resulted in being connected to joint laxity, as underlined by the correlation between the graft T2 signal and the temporal evolution of knee function. Conclusions: This preliminary study represents a step forward in assessing the effects of ACL graft maturation on knee biomechanics, and vice versa. The presented integrated framework underlines the possibility to quantitatively assess the impact of ACL reconstruction on trauma recovery and cartilage homeostasis. Moreover, the reported findings—despite the preliminary nature of the clinical impacts—evidence the possibility of monitoring the surgery outcomes using a multi-parametric prognostic investigation tool.

## 1. Introduction

Anterior cruciate ligament (ACL) tears represent one of the most critical issues affecting the human knee [1]; in fact, ACL injuries significantly impair joint biomechanics by increasing knee laxity in both the frontal and transverse plane [2,3]. Common reconstructive techniques imply the substitution of the injured ligament by using a biological graft to restore knee function with good clinical outcomes [4].

Even though this kind of treatment has substantially been improved over time, the optimal approach is still a challenging issue in orthopedic research and clinical practice [5]. Sub-optimal reconstructions predispose patients to both re-injuries and early development of osteoarthritis (OA) [6]. Regarding the latter, long-term studies have demonstrated that around 50% of patients undergoing ACL reconstruction still develop early cartilage degeneration, primarily due to residual joint laxities [7,8].

To assess the knee status after ACL tear and following reconstruction, one of the most widely used, non-invasive, and predictive method is indeed represented by magnetic resonance imaging (MRI) [9,10,11], which allows a structural integrity observation of both cartilage and graft. Several sequences sensitive to biochemical and compositional changes within soft tissues were developed to identify specific conditions related to alterations of the joint homeostasis including collagen fiber network disorders, water content variations, and glycosaminoglycan depletion [12]. Among the most used sequences, the transverse relaxation time, namely T2, was reported to be strongly influenced by the presence of free water molecules [13]: in fact, T2 signal relates to the speed by which nuclei lose phase coherence following excitation, resulting in an exponential decay of transverse magnetization and MR signal. Additionally, the T2 signal is also dependent on both the collagen content and fiber orientation, the loss of type II collagen [14,15], and proteoglycan content [16]. 

T2 sequence was widely used in the assessment of cartilage status after ACL reconstruction and, more recently, extended to the evaluation of ACL graft maturation [11,17,18]. After implantation, ACL graft is subjected to a sequential remodeling process named ligamentization, during which architecture and composition, along with fibril alignment and vascularity, undergo a drastic change [19]. Understanding the biological processes occurring during such a phase is of paramount importance to obtain a successful reconstruction [20]. Besides graft maturation, an alteration of knee cartilage tissue can be observed after both injury and ACL reconstruction because of alterations to joint biomechanics [21]. After ACL tearing, a shift in the cartilage contact regions is in fact evidenced, producing increased loads in those areas that generally perform very limited bearing tasks; vice versa, a reduced loading condition can be further noticed in portions of tissue that used to be subjected to frequent loads [22]. These variations may cause articular surface damage and increase fibrillation of the collagen network [23]. Even after ACL reconstruction, residual abnormal joint biomechanics could be present and lead to an early degeneration of the cartilaginous tissue [24]. Moreover, considering the complex scenario characterizing the traumatic injury of ACL, and hence its reconstruction, it is crucial to determine the relationship between knee function, which could be clinically assessed through joint laxity, and the conditions of the main tissues present within the joint. Although these evaluations were individually performed, no reports appear to relate knee laxity with the biological status of the soft tissues primarily involved in the healing process after ACL trauma and the following the reconstruction (i.e., the graft and the articular cartilage). 

The present work aimed at providing a multi-parametric framework capable of supporting a quantitative assessment of the main outcomes related to knee function and soft tissue condition after traumatic rupture of ACL and following its reconstruction. This integrated evaluation—performed by using knee laxity and quantitative MRI T2 mapping analysis applied to both the ligamentization process and the cartilage status—is able to properly define diagnostic and prognostic key parameters that can be used to guide the surgery and the following rehabilitation process, since the alterations in the soft tissues are indeed mediated by the changes in the knee biomechanics. A preliminary pilot study was specifically performed to verify these hypotheses.

## 2. Methods

### 2.1. Patients Enrolment

The present study was based on a prospective randomized control study approved by the local ethical committee (ID: 0013262-19/04/2013 Clinical Trial Gov ID: NCT05088278). The involved subjects were instructed about the analyses intended and voluntarily signed the informed consent.

Inclusion criteria were identified as follows: (i) age between 16 and 50 years; (ii) concomitant absence of meniscal or chondral injuries, which were assessed through MRI and verified during arthroscopy; (iii) skeletal maturity, evaluated through radiological assessment of the proximal tibial grow plate [25]. Patients were specifically included in this analysis if they underwent surgical ACL reconstruction performed through one of the implemented single bundle (SB) approaches; these reconstruction techniques previously demonstrated no difference in terms of reducing both the antero-posterior translation at 90° knee flexion, and the internal–external rotation at 30° knee flexion [3]; all surgeries were performed by using semitendinosus and gracilis tendon autografts [18,26]. Exclusion criteria were: (i) acute ACL lesion [26]; (ii) prior knee surgery; and (iii) risk factors for osteoarthritis or other forms of arthritis. Patients (N = 13, two females and 11 males, aged 21 ± 4.0 years) who matched the reported criteria were identified and included in this study.

Functionality and imaging assessments (see next sections) were performed before the reconstruction at four months (FU4) and at 18 months (FU18) follow-up. A temporal distance of four months from the surgery was considered a reliable time to start assessing the remodeling process of the tissues, whereas 18 months were considered enough to reach a final biomechanically coherent status.

In order to assess the T2 relaxation time on both the ACL and the cartilaginous tissue in a control group, a further cohort of five non pathological knees was investigated.

### 2.2. Knee Laxity Assessment

The clinical assessment of the antero-posterior laxity was performed by means of two different methods. A KT-1000 arthrometer (MEDmetric Corp., San Diego, CA, USA) was used to quantify the anterior displacement of the tibia with respect to the femur during the Lachman test by applying both a 134 N force (KT-Load) and a manual-maximum force (KT-Man) [27]. A rolimeter (Aircast Europa, Neubeuern, Germany) was also used to assess the joint antero-posterior laxity during the drawer test [28]. Finally, the Kira (Orthokey, Italy) system, embedding an inertial sensor, was used to obtain quantitative information about the dynamic joint laxity during pivot shift maneuver [29]. Contralateral and ipsilateral joint were assessed in each patient.

### 2.3. MRI Analysis

MR scans were acquired before surgery and at four and 18 months after the reconstruction; following this protocol, we were able to assess the characteristic T2 signal of both the injured ACL (in the pre-operative stage), the graft, and the articular cartilage over time. 

All scans were performed with a 1.5 T whole-body scanner (General Electric Medical Systems) by using a dedicated eight-channel Phased Array Knee Coil. A multiple time-echo (TE) fast spin sagittal pulse sequence (SAG T2) was used for T2 relaxometry analysis. The T2 relaxation imaging was performed using spin echo datasets with the following parameters: repetition time (TR) = (1000 ÷ 1100) ms; echo time (TE) = (8 ÷ 72) ms; field of view (FOV) = 160 mm; flip angle (FA) = 90°; slice thickness = 3 mm, slice spacing = 3.5 mm, matrix = 512 × 512, band width (BW) = 122 Hz/pixel. MR sequences were transferred to a workstation for the offline quantification of the T2 signal. A single investigator performed all morphometric segmentations through a dedicated and previously adopted software [30]. 

Concerning the evaluation of the T2 signal for the injured ligament, healthy ligament, and graft, the slice of each sagittal sequence showing the clearest image of Blumensaat’s line was selected [31,32]. From this perspective, the use of a single section to acquire the T2 signal allows for the exclusion of undesirable factors such as magnetization transfer and crosstalk; these issues might indeed occur if a multi-section acquisition is used [33,34]. After obtaining the T2 map by applying the Levenberg–Marquardt algorithm [35], the contours of the intra-articular portion of the graft/ACL were manually outlined on the mask obtained by overlaying the starting slice to the corresponding region in each T2 map image [36] (Figure 1). This procedure allowed us to evaluate the T2 signal of the purely tendinous/ligamentous portion, achieving substantial segregation from surrounding synovial tissue, which usually occurs throughout the course of graft maturation [37]. 

Concerning the assessment of the knee cartilage status via T2 signal, and according to relevant scientific literature, the regions of interest (ROIs) able to significantly highlight any onset of early cartilage degeneration are the posterior lateral tibial plateau (pLTP), the central medial tibial plateau (cMTP), and the central medial femoral condyle (cMFC) (Figure 2) [6,38]. Each sequence was therefore divided on a slice-by-slice basis and by anatomical markers, specifically by the menisci of the knee [39]. The mid-sagittal slice was selected for the evaluation of the T2 signal in the above-mentioned ROIs [36], specifically by following the same procedure previously reported for the assessment of ligaments and graft.

As previously highlighted, the T2 measurements enabled us to elucidate advances in the ligamentization process and/or alteration of the cartilage status. Aiming to highlight any possible relation between the status of the soft tissues and the healing process, the T2 signals of the injured ACL, the graft, and the cartilaginous tissue in the patients were compared with the results obtained from both the control group and the corresponding scientific literature. 

### 2.4. Statistical Analysis

The Kolmogorov–Smirnov test was preliminary performed on the whole dataset to examine the normality of the distribution of the data. Two-sided Wilcoxon rank sum test (*p* = 0.05) was used to assess differences between “treated” and “control” groups in terms of age and body mass index (BMI). Two-sided Wilcoxon rank sum test (*p* = 0.05) was used to investigate statistical significance concerning prospective changes in joint laxity, and in both graft and cartilage status; moreover, the same test was used to statistically investigate any difference in T2 signal (i) between healthy ACL, graft and injured ACL, and (ii) between cartilage ROIs in healthy and treated joints. Regarding the joint laxity, we considered both the side-to-side difference and the absolute values in the reconstructed joint, evaluated before the reconstruction and at FU4 and FU18; furthermore, we analyzed the changes during the follow-up phase (i.e., ΔFU–), thereby tracking the evolution of joint function after the surgery. Concerning the biological status of the graft and cartilaginous tissue, the T2 signal was investigated among the same assessment phases, specifically considering their difference between FU4 and FU18, aiming at highlighting any pathophysiological change due to the reconstruction.

Finally, assuming that there was a relationship between graft maturation, knee stability, and cartilage status [40], a correlation analysis was performed between joint laxity data and both graft and cartilage T2 signal data by means of the Spearman’s method (Spearman’s rank correlation coefficients; p: testing the hypothesis of no correlation against the alternative hypothesis of a nonzero correlation; and R: measure of the strength and direction of the relationships between two ranked variables). All the statistical analyses were performed using MATLAB (MathWorks, Natick, MA, USA).

## 3. Results

Demographic characteristics of the recruited subjects are reported in Table 1.

No statistically significant differences were found in terms of BMI between pre-operative and follow-up times. Furthermore, no differences were noticed between the treated and control groups in terms of age and BMI, with the latter considered at each evaluation time.

### 3.1. Knee Laxity 

No statistically significant correlation was found between joint laxity (i.e., side-to-side, absolute values, and ΔFU–) and the subjects’ demographic data (i.e., age, BMI, and time injury-to-surgery). Side-to-side and absolute knee laxity data at each evaluated time point are reported in Figure 3 and Table 2, respectively (see also Appendix A).

A significant decrease in the side-to-side laxity (assessed through KT-Load, KT-Man, and by rolimeter) was highlighted between pre-op and FU4 (*p* < 0.05). The rolimeter indicated a further laxity decrement between pre-op and FU18 (*p* < 0.01). No significant differences were recorded by Kira, although a clear decreasing trend was noticeable.

Treated joint in the pre-operative phase showed a statistical higher laxity compared to the contralateral knee. Statistical difference between treated and contralateral knee was reported at FU4 and FU18, both by KT-Load and KT-Man. Statistical difference in joint laxity of the treated knee was highlighted by KT-Load and rolimeter between pre-operative phase and FU18. 

### 3.2. ACL, Graft, and Cartilage T2 Signals

T2 signals of the injured and healthy (control) ACL, and the graft at FU4 and at FU18 are reported in Table 3 (see also Appendix A).

No statistically significant correlation was found between injured ACL T2 signal and the subjects’ demographic data (age, BMI, and time injury-to-surgery). Concerning healthy ACL evaluation, statistical differences were identified with respect to the graft, evaluated both at FU4 and at FU18 (*p* < 0.01). Regarding injured ACL, statistical differences were identified with respect to the graft, evaluated both at FU4 and at FU18 (*p* < 0.01). No difference was found between injured and healthy ACL T2 relaxation time. 

T2 signals of the evaluated knee cartilage ROIs for treated subjects at pre-operative, FU4, and FU18 and for the control group are reported in Table 4 (see also Appendix A).

No correlation was present between the cartilage T2 signal and the subjects’ demographic data. Regardless of the ROI, no statistically significant differences in cartilage T2 were highlighted between pre-op and both FU4 and FU18; moreover, no statistically significant differences were reported in terms of T2 signal between the control and treated subjects, regardless of the ROI and the follow-up times. 

### 3.3. Relationship between Cartilage and Graft T2 Values

A negative fair correlation between the T2 value of pLTP and graft at FU4 was highlighted between the cartilage and graft status (*p* < 0.05, R = −0.5055; see Appendix A), in other words, the higher the pLTP T2 signal at FU4, the lower the graft T2 value at the same time-point. Moreover, a fair–moderate correlation was found between the T2 value of the graft at the FU4 and cMTP ΔFU (*p* < 0.05, R = 0.6264; see Appendix A), in other words, the higher the graft T2 signal at FU4, the higher the cMTP T2 value at FU18.

### 3.4. Relationship between Cartilage T2 Values and Knee Laxity 

A relationship between cartilage status and joint functionality was revealed as early as the pre-operative phase, in which a moderate correlation was found between the pLTP T2 value and absolute knee laxity, assessed by using the rolimeter (*p* < 0.05, R = 0.6152; see Appendix A), in other words., the greater the laxity, the higher the pLTP T2 signal.

At FU4, a fair–moderate correlation was found between the cMTP T2 value and absolute joint laxity, assessed by both the KT-Load and rolimeter (*p* < 0.05, R = 0.5739 and 0.5810, respectively; see Appendix A), in other words, the greater the laxity, the higher the cMTP T2 signal. 

At FU18, focusing on pLTP, a moderate correlation was found between ΔFU T2 value and side-to-side joint laxity evaluated by the rolimeter (*p* < 0.01, R = 0.8345; see Appendix A), in other words the greater the laxity, the higher the pLTP T2 signal. 

### 3.5. Relationship between Graft T2 Signal and Knee Laxity 

Concerning the evaluation of a relationship between graft maturation and joint functionality, and focusing on side-to-side laxity, correlations were found at FU4 between the graft T2 value and laxity assessed by using both KT-Load and KT-Man (*p* < 0.05, R = −0.5125 and −0.6714, respectively; see Appendix A). They indicate that the greater the laxity at FU4, the higher the graft T2 signal.

Moving on to the absolute values of joint laxity, Spearman’s analysis revealed a moderate correlation between the graft T2 value and knee laxity at FU4 assessed through KT-Load (*p* < 0.05, R = −0.5878), KT-Man (*p* < 0.01, R = −0.6944), and rolimeter (*p* < 0.05, R = −0.5924): the greater the laxity, the lower the graft T2 signal (see Appendix A, respectively). Moreover, a moderate correlation between T2 value at FU4 and joint antero-posterior translation at FU18 evaluated by rolimeter (*p* < 0.05, R = −0.6005; see Appendix A) was also highlighted, indicating the lower the graft T2 signal, the higher the laxity.

## 4. Discussion

The present pilot study aimed at promoting the use of a comprehensive methodological multi-parametric approach capable of quantifying the relationships between the status of the soft tissues within the knee joint and its biomechanical condition, which are indeed important in the assessment of both the trauma and the effects of the reconstruction.

In particular, the status of the graft, here assessed through quantitative MRI T2 mapping, showed interesting relationships with both knee cartilage status and joint laxity. In fact, concerning the link between graft and cartilage status, a negative correlation was specifically found between the graft and pLTP T2 value assessed at FU4. This first evidence may suggest that the status of these tissues is related within the knee joint. Moreover, graft maturation resulted in being connected to joint laxity, as underlined by the result obtained through the correlation analysis performed between the graft T2 value at FU4 and the evolution of knee laxity during the follow-up period. This finding showed that a higher T2 signal for the graft—and therefore its changes in microstructure/composition due to the ligamentization process—is related to a reduced joint laxity; furthermore, the condition of the graft might predict the laxity of the knee.

ACL injury represents a common orthopedic issue that could lead to an early development of post-traumatic osteoarthritis, if not properly treated [40]. From this perspective, reconstruction parameters including graft choice, placement, fixation, and tension as well as the general laxity could definitely affect the knee biomechanics, possibly influencing the outcome of the surgery itself [41]. Moreover, post-operative inflammation may damage synovial stem cells and lead to a compromised joint environment, thus affecting the ability of the tissues to heal [42]. Therefore, there is a compelling need to improve diagnostic and prognostic techniques aiming to detect the processes occurring at the joint level after substituting an injured ACL. Accordingly, assessing the status of the graft during the post-operative phase, and its connection with both joint stability and articular cartilage status is a key point in determining the appropriate timing for a return to sporting activity and in preventing re-injury [33]. Thus, a full insight into the relationship between biological processes and joint function represents a fundamental step in properly evaluating the outcome of the surgical treatments.

Starting from these premises, hereinafter, we report several considerations that we can draw from the pilot study we performed by applying this multi-parametric approach.

### 4.1. Knee Laxity

Concerning joint function, the applied reconstructive techniques succeeded in significantly reducing the knee laxity considering both the static displacements and the dynamic pivot shift as already reported in the literature [3]. Abnormal joint laxity is indeed a risk factor for subjective instability, meniscal injuries, and early onset of OA [43]. Although most reports on residual laxity after ACL reconstruction are limited to short-term follow-up, a side-to-side difference <5 mm after ACL reconstruction is generally accepted as a successful result [44]. Nevertheless, joint laxity following ACL reconstruction is influenced by the biomechanical and histological properties of the graft [45]. Thus, to argue properly about the success of the reconstruction, it is mandatory to consider the relationship between knee functional and pathophysiological outcomes. 

### 4.2. ACL T2 Signal 

Considering T2 mapping, it is noteworthy to mention that the T2 signal is primarily related to the presence of free water molecules and, to a lesser extent, to collagen type and content, fiber orientation, and proteoglycans [13,14,15,16]. Consequently, a variation in water content has a greater effect on T2 signal when compared to the changes concerning the other main components. In the presence of an acutely injured ACL, a progressive decrease in the joint fluids, and thus of the water content, within the articular tissues is reported [46]. Moreover, an injured ACL presents a lower amount of collagen and proteoglycans when compared to the healthy one [47]. Accordingly, it is reasonable to assume that a decrease in these components as a consequence of a trauma should produce a reduction in the T2 signal. In this study, the lack of statistically significant difference between healthy and pathological ACL could hence be ascribable to the time when we acquired the T2 signal (i.e., 3.9 ± 2.3 months after the injury). We can hypothesize that this time, which corresponds to a sub-acute or chronic condition, seems to be sufficient to restore a stable status in the only terms of composition and surrounding biological environment. It is worth underlying that our data are also in agreement with the values reported by Schmitz et al. [31].

### 4.3. Graft T2 Signal 

Focusing on the tendon graft, we measured a significantly lower value of the T2 signal when comparted to the healthy ACL; this finding is also in agreement with previous studies [48,49]. In particular, Chu et al. reported a sensible reduction in T2 signal in tendons with respect to healthy ACLs. Despite ligaments and tendons having a similar composition, the first present a higher content of water and a slightly lower amount of collagen [50,51]. Moreover, these tissues also differ in the orientation and organization of the collagen structure [52]; while ACLs—and ligaments in general—are complex band-like structures of dense connective tissue, tendons present a well-defined collagen structure, in which the fibers assume a preferential orientation [53]. Therefore, when comparing T2 signal in tendons and ligaments, it is possible to suggest that the difference in their values is due to both a different arrangement of the collagen structure and a distinct water content. In order to understand the changes in the graft T2 signal during the follow-up, it is fundamental to focus on the biological modifications occurring in such a period. Once implanted within the joint, tendon grafts should undergo three characteristic stages of adaptation: remodeling, proliferation, and ligamentization [54], graphically summarized in Figure 4, which also reports a qualitative estimation of the theoretical progress of the T2 signal.

Remodeling phase generally lasts up to four months [55], reporting irregular collagen orientation, graft necrosis, hypo-cellularity, and a sensitive decrease in vascularization [54,55,56]. Consequently, the mechanical strength of the graft becomes significantly lower in this period [20]. Moving to the proliferation phase, which lasts approximately 12 months after the reconstruction, it is characterized by an intensive revascularization process [57,58]; at this stage, changes in extracellular matrix and a significant decrease in collagen fibril density are usually detectable [20]. Finally, the ligamentization phase involves the continuous remodeling of the graft toward the biological and mechanical properties of the native ACL. It is noteworthy to mention that a clear endpoint of the ligamentization process can hardly be defined, since changes in the graft might occur years after the reconstruction [54]. Therefore, although it is possible to reliably define a relationship between the biological changes and T2 signal during the initial phases of the adaptation process, it is quite difficult to identify a similar behavior in the last stage.

In this study, we performed the assessment at four (FU4) and 18 (FU18) months after the surgery; these times should specifically correspond to the period between the remodeling and proliferation phase, and to an ongoing stage of graft ligamentization, respectively. Concerning the graft, the temporal evolution of the T2 signal we measured during the follow-up period is coherent to the information highlighted by the literature [11,18]; in particular, we found lower average values and reduced standard deviation at FU18 compared to the value acquired at FU4 (see Table 3, previously reported, and Figure 5).

In this frame, a decrease in T2 signal between FU4 and FU18 may suggest that the graft is still in the proliferation phase, and a stable ligamentization of the tissue has yet to take place. Accordingly, an increase in the temporal duration of the proliferation phase, besides the above-mentioned period, can be suggested. Moreover, considering the peculiar values of T2 signal for the healthy ACL, as reported by the literature, we can also suggest that the T2 values should increase during the post-operative phase when the graft aims at reaching the peculiar composition and structure of the native uninjured ACL.

### 4.4. Cartilage T2 Signal 

Regarding the cartilage tissue, the T2 signal is reported to be inversely correlated with both collagen network organization and structure and, furthermore, it is directly correlated with the content of free water [59]. When the extracellular matrix of the articular cartilage is compromised i.e., when there are early biochemical processes related to the OA onset), water moves more freely within the cartilage, prolonging the T2 relaxation time [60]. Moreover, cartilage T2 signal was demonstrated to be able to predict patient-reported outcomes after ACL reconstruction, highlighting a link between biomechanics and OA development/progression [61,62]. Accordingly, a quantitative evaluation of the changes in cartilage T2 signal after the surgery may provide additional insight into cartilage matrix properties, which are indicative of the injury and degeneration [10]. As reported above, it is noteworthy to mention that changes in the homeostasis of the articular cartilage relate to the alteration of the joint biomechanics, considering both the ones due to ACL tear and those linked to a possible not-fully-restored joint function after the reconstruction. Although the presence of different MRI protocols may explain some discrepancies, our T2 values are comparable to the ranges reported in ACL-related studies [6,62,63,64,65,66,67]. Nevertheless, as previously highlighted, it is mandatory to evaluate the progression of T2 signal during the follow up period, to correctly understand the degenerative processes affecting joint cartilage [10,36]. A progressive increase in cartilage T2 value was previously noted in patients affected by OA [36]. Moreover, an increase in cartilage T2 signal was already detected in subjects after ACL reconstruction, particularly considering medial femur and lateral tibia [66]. In contrast, in this study, we were not able to find any statistically significant change in cartilage T2 signal, neither between pre-operative and follow-up phases, nor between FU4 and FU18 periods. We hypothesized that the first lack of difference could be ascribable to the fact that subjects did not overstress the knee joint after the injury, thus, the altered joint biomechanics did not produce significant changes in the cartilage homeostasis during the injury-to-surgery phase. This hypothesis is further supported by the absence of a statistically significant difference in T2 signal between healthy and pre-op cartilage. Concerning the lack of difference during the follow-up, it is possible to hypothesize that, due to a proper restoration of the knee function, no signs of early degeneration were recorded, at least if we consider the only T2 analysis. Although an ACL-deficient knee control group may better support this hypothesis, Potter et al. [68], however, reported an increased risk of cartilage degeneration in non-surgical patients compared to the surgically treated ones due to the lack of biomechanics restoration, thus underlying the beneficial effects of ACL reconstruction on cartilage tissue homeostasis.

### 4.5. Relationship between Graft T2 Signal and Knee Laxity 

Analyzing the scientific literature, few studies have focused on the connection between graft ligamentization and knee function [11,18,69]. Niki et al. investigated the post-operative outcomes of ACL reconstruction through T2 signal and laxity measurements [11]. They demonstrated the capability of quantitative MRI for predicting knee function due to positive correlations found between the T2 signal measured at 12 months and the antero-posterior laxity quantified at both 24 and 48 months. However, the authors reported a progressive increase in knee laxity during the follow-up, which probably suggests a partial failure of the reconstruction in restoring the normal joint function. Lansdown et al. evaluated the longitudinal progression of the graft characteristics following ACL reconstruction by using advanced quantitative imaging measurements [18]. They found a negative correlation between the graft T2 signal and patient-reported outcomes (in particular KOOS sub-scores) both evaluated at 24 months follow-up. However, although these measurements represent reliable tools in subjectively assessing the reconstruction success, they do not provide information on the functional status of the knee. In the present study, T2 signal was evaluated both at the very end of the remodeling phase and at an early stage of ligamentization. Negative correlations were found between the T2 signal of the graft and knee laxity at FU4; moreover, a moderate correlation between graft T2 value at FU4 and joint laxity at FU18 was also highlighted. As reported above, substantial differences occurred to the graft when moving from one phase to another. Thus, before considering the feasibility of MRI in evaluating and predicting knee function, it is crucial to interpret in a correct manner the trend extrapolated from the T2 analysis in the follow-up. In this perspective, Niki et al. [11] and Lansdown et al. [18] inferred that the graft remodeling process was similar to the one reported for the cartilaginous tissue, in which an elevated T2 signal can be associated with a degenerative or, more in general, to a not-healthy condition. Nevertheless, ligaments and tendons show similarities with respect to cartilage in terms of composition, but they strongly differ in terms of internal structure [51,52]. Therefore, they need an ad hoc evaluation: results evidenced both the punctual and the predictive relationship between the graft T2 signal at FU4 and the knee stability at FU4 and FU18. Considering the above reported progression in graft T2 signal during its maturation, it is possible to argue that a graft in a more advanced phase of ligamentization (i.e., characterized by an elevated T2 value) could better restore knee laxity after ACL reconstruction. 

### 4.6. Relationship between Cartilage T2 Signal and Knee Laxity 

Considering the relationship between cartilage status and knee laxity, here supported by the fairly good Spearman’s correlation found between ΔFU T2 value and side-to-side joint laxity (*p* < 0.01, R = 0.8345), the results suggest that reducing joint laxity could avoid an increase in the cartilage T2 signal, in particular by focusing on the tibial articular surface. Despite a relationship between gait variables and cartilage status already highlighted in the literature [61,67], there is a lack of evidence on the presence of a correlation between knee laxity and cartilage T2 signal. The main goal of reconstructing ACL through tendinous substitutes is restoring knee function, thus reducing the possibility of reinjury and further damage to the knee as well as to facilitate sports participation [45]. Residual laxity after surgery was proven to have an effect on long-term joint biomechanics, even leading to the graft failure [44]. Therefore, considering both the above-mentioned relationship and the main findings reported by Potter et al. [68], it is reasonable to argue that the lower the knee laxity after ACL reconstruction, the less the probability of progressive degeneration of the cartilaginous tissue. 

### 4.7. Relationship between Cartilage and Graft T2 Signal 

Concerning the relationship between the graft and cartilage status, a negative correlation was here highlighted in terms of the graft and pLTP cartilage T2 signal. Despite previous studies focusing on the association between increased knee cartilage T2 signal and various stages of OA [70,71,72,73], this work first evidenced a possible link between the graft and knee cartilage status following ACL reconstruction. The found correlation allows us to argue that a more elevated T2 graft signal, a sign of an advanced ligamentization phase also related to a better restoration of the joint laxity, can be associated with an articular cartilage that is not affected by an early degenerative process (at least when considering the analysis of the T2 signal). This correlation closes the loop in evaluating the effects of ACL reconstruction on knee biomechanics; therefore, the graft maturation seems to be the driving factor in restoring joint function by influencing complex aspects of the reconstruction and avoiding cartilage degeneration.

Finally, although this work aimed at presenting an overall methodological framework, some caveats concerning several points we previously discussed must be underlined.

The small number of patients represents one of the major limitations of this study. Nevertheless, the purpose of this research was to develop an integrated multi-parametric approach aiming at properly investigating the outcomes of ACL surgery reconstruction within a coherent clinical framework, and not to provide general clinical conclusions. In any case, we were able to provide several hints about the possible relationships between knee tissue status and joint function, suggesting that the graft maturation plays a crucial role in driving the ACL reconstruction outcomes.

Focusing on the control group, a cohort of five non-pathological knees was reported here as a reference for the analysis. Although the contralateral joints would have represented the ideal choice, it is noteworthy to mention that no statistically significant differences between the treated and control groups were reported in terms of BMI and age; moreover, also considering the pilot nature of the proposed methodological framework, the reported considerations, particularly with regard to T2 signal analysis, were also supported by the scientific literature.

Regarding MRI analysis, a baseline assessment would have allowed a better evaluation of the effects of an altered joint condition on the homeostasis of the cartilage tissue, and should be included in a real clinical study. However, the absence of a statistically significant difference between non-pathological and pre-operative cartilage in the T2 signal suggests that the involved patients did not overstress the knee joint after the injury, thus avoiding an alteration of the joint biomechanics that would lead to significant changes in cartilage homeostasis during the injury-to-surgery period. 

Focusing on the ligamentization process, at 18 months, the graft did not seem to be, with a reasonable confidence, in a fully remodeled condition. Longer follow-ups are required to reliably conclude on the success of the ACL reconstruction (i.e., when the graft reaches the T2 signal of the healthy ACL). 

Focusing on the evolution of the cartilage status during the follow-up periods, a control group including ACL-deficient knees could have better supported the hypothesis that restoring joint kinematics is beneficial to cartilage homeostasis. Actually, although we can argue about the ethics of enrolling such a control group, the scientific literature has reported an increased risk of cartilage degeneration in non-surgical patients compared to the surgically treated ones, mainly due to the lack in restoring joint biomechanics [68].

Notwithstanding the limitations mainly related to the pilot nature of the study, the proposed framework allowed us to provide a step forward in assessing the effects of the graft maturation process on knee biomechanics, which seems to have direct effects on the joint cartilage conditions. In general, the achieved findings, due to the relationships existing between the graft T2 signal and both knee laxity and cartilage status, suggest the possibility of monitoring both the biological and functional outcomes during the follow-up phase by using a single clinical non-invasive investigation and thus improving the overall prognostic capability.

The developed multi-parametric framework should also be considered in properly defining the optimal ACL reconstructive technique, aiming at simultaneously acting in both restoring joint biomechanics and improving graft ligamentization, avoiding the rise in any early OA-related degenerative process.

## Figures and Tables

**Figure 1 life-11-01383-f001:**
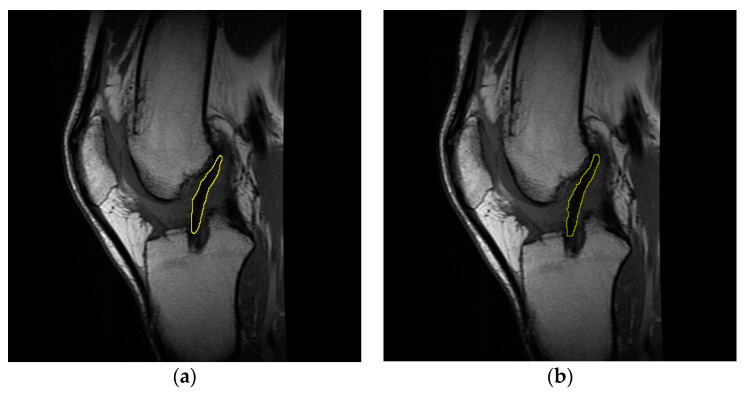
Representative sagittal images of the graft intra-articular portion, manually outlined on the starting slice (**a**) and on the mask obtained by overlaying the latter to the T2 map (**b**). Data concern subject #6 at four months follow-up.

**Figure 2 life-11-01383-f002:**
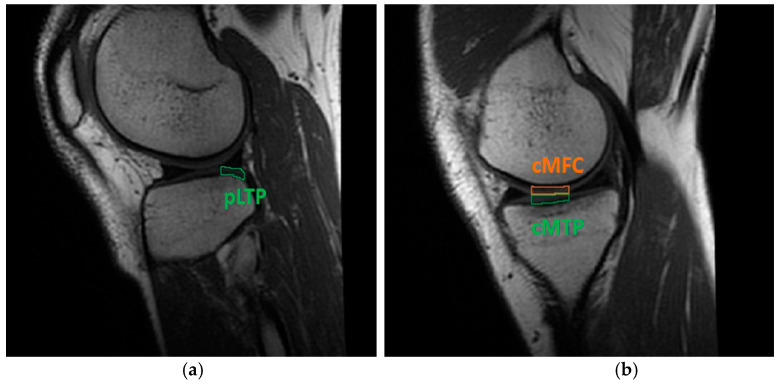
Cartilage ROIs evaluated through T2 mapping software: (**a**) pLTP portion in the lateral knee compartment; (**b**) cMFC and cMTP regions in the medial compartment. Data concern subject #6 at four months follow-up.

**Figure 3 life-11-01383-f003:**
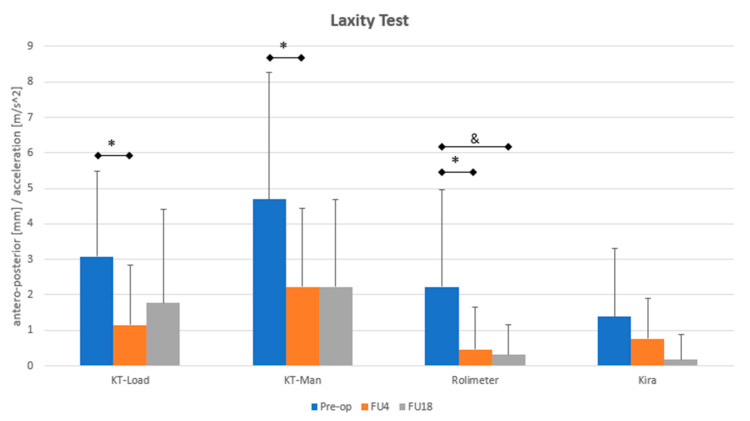
Side-to-side joint laxity assessment of the treated subjects at each evaluation time point (pre-op, FU4, and FU18). ***** means statistical difference (*p* < 0.05) between Pre-op and FU4. **&** indicates statistical difference (*p* < 0.01) between pre-op and FU18.

**Figure 4 life-11-01383-f004:**
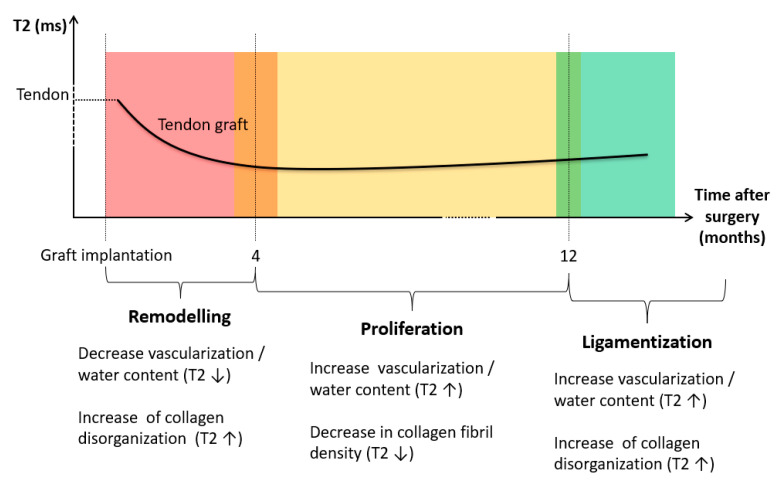
Biological changes occurring to the tendinous graft after the implantation, and consequent theoretical modifications produced at the level of the T2 signal.

**Figure 5 life-11-01383-f005:**
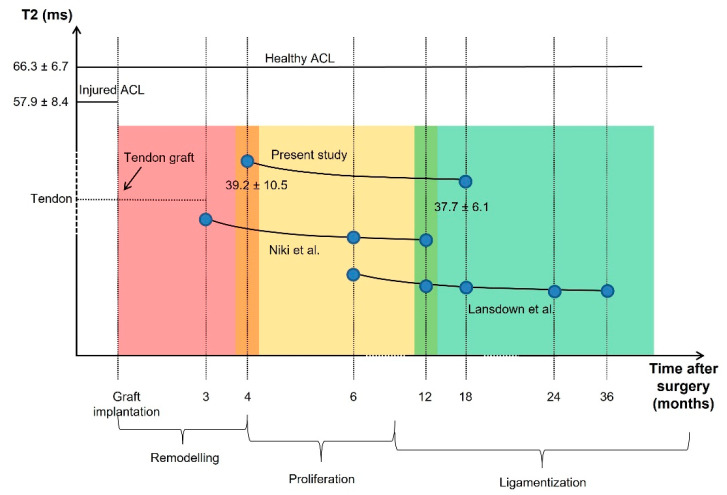
The trend of the graft T2 signal assessed in the present study is shown. Blue points represent the month at which graft T2 relaxation times were obtained. Moreover, the graft T2 signals provided by Niki et al. [11] and by Lansdown et al. [18] are reported.

**Table 1 life-11-01383-t001:** Demographic data (age at surgery, time injury-to-surgery, BMI) of the treated patients and control group.

**Age at Surgery (Years)**	Treated	21.2 ± 4.1
	Control	28.8 ± 14.9
**Time Injury-to-Surgery (Months)**		3.9 ± 2.3
**BMI (kg/m^2^)**	Pre-op	23.5 ± 2.8
	FU4	23.0 ± 2.6
	FU18	23.0 ± 2.6
	Control	22.4 ± 3.67

**Table 2 life-11-01383-t002:** Absolute values of joint laxity of both treated and contralateral (Cont) joint, at each evaluation time point (Pre-op, FU4, and FU18). $ means statistical difference (*p* < 0.01) between Pre-op and FU18. * indicates statistical difference (*p* < 0.01) between treated and contralateral joint. @ means statistical difference (*p* < 0.05) between treated and contralateral joint.

	KT-Load (mm)		KT-Man (mm)		Rolimeter (mm)		Kira (m/s^2^)	
	Treated	Cont	Treated	Cont	Treated	Cont	Treated	Cont
Pre-op	8.5 ± 2.0 $*	5.5 ± 2.1 *	12.1 ± 2.7 *	7.4 ± 2.0 *	6.9 ± 1.7 $ @	4.7 ± 1.9 @	5.4 ± 1.9	3.6 ± 1.1
FU4	7.1 ± 2.0	5.9 ± 2.1	10.9 ± 2.3 @	8.7 ± 2.1 @	5.6 ± 2.0	5.2 ± 2.6	3.1 ± 0.8	3.3 ± 0.8
FU18	6.9 ± 2.5 $ @	5.1 ± 1.6 @	10.9 ± 3.1 @	8.6 ± 2.0 @	4.3 ± 1.7 $	4.0 ± 1.4	4.0 ± 1.9	3.8 ± 1.8

**Table 3 life-11-01383-t003:** T2 of the injured, healthy (control) ACL, and graft at FU4 and FU18 of the treated subjects. $ indicates statistical difference (*p* < 0.01) between injured ACL and graft considering T2 signal. £ means statistical difference (*p* < 0.01) occurred in the T2 between healthy ACL and graft.

**T2** (**ms**)	Injured ACL	Pre-op	57.9 ± 8.4 $
	Healthy ACL	Control	66.3 ± 6.7 £
	Graft	FU4	39.2 ± 10.5 $ £
	Graft	FU18	37.7 ± 6.1 $ £

**Table 4 life-11-01383-t004:** T2 signal of the posterior lateral tibial plateau (pLTP), central medial tibial plateau (cMTP), and central medial femoral condyle (cMFC) achieved at each evaluation time point (Pre-op, FU4, and FU18) for the treated subjects (N = 13) and for the control group (N = 5).

**T2** (**ms**)		**pLTP**	**cMTP**	**cMFC**
	Pre-op	42.2 ± 3.9	37.1 ± 3.4	35.7 ± 5.2
	FU4	42.2 ± 4.2	36.8 ± 3.4	33.7 ± 4.0
	FU18	42.7 ± 3.7	35.4 ± 3.1	35.0 ± 4.3
	Control	38.3 ± 3.4	38.0 ± 5.4	33.9 ± 4.5

## Data Availability

The data presented in this study are available in the Appendix A.

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
