# Peer review of "A Comprehensive Framework to Evaluate the Effects of Anterior Cruciate Ligament Injury and Reconstruction on Graft and Cartilage Status through the Analysis of MRI T2 Relaxation Time and Knee Laxity: A Pilot Study"

_life, 2021, doi:10.3390/life11121383_

Round 1

Reviewer 1 Report

A relatively small number of patients, with only 13 individuals - statistic may be impaired. 

Line 102 how do you define “chronic ACL lesion”. In other words, patients included in the study were how long after the initial trauma? 

Line 117 As this is a prospective study, please explain why these particular time points were chosen: 4 and 18 months after the surgery? 

Line 147 why ROI in this particular area was chosen? Please explain.

Line 212 It would be good to compare the data to the T2 signal of either intact ACL or the graft immediately after reconstruction or other healthy intraarticular ligaments, ex PCL. 

Line 391 With no control group (evaluating cartilage T2 signal changes with ACL deficient knee at 4 and 18 months FU), one should not hypothesize the influence of restoration of joint kinematics on cartilage status. Furthermore, pre-op MRI was performed in acutely injured knees – did it impact cartilage T2 signal? Maybe comparing with the contralateral knee values would be beneficial? 

Line 430 There is no evidence in the present study allowing for that conclusion. There is no control group showing an increase in cartilage T2 signal in ACL deficient knees at 18 months FU. 

Please include some of the more recent studies evaluating ACL graft maturation, ex: 

Zdanowicz, U.; Ciszkowska-Łysoń, B.; Paśnik, M.; Drwięga, M.; Ratajczak, K.; Fulawka, K.; Lee, Y.C.; Śmigielski, R. Evaluation of ACL Graft Remodeling and Prediction of Graft Insufficiency in Sequenced MRI—Two-Year Follow-Up. Appl. Sci. 2021, 11, 5278. https://doi.org/10.3390/app11115278

Reviewer 2 Report

This study investigated the framework for evaluating the effects of ACL injury and reconstruction on cartilage status through MRI T2 relaxation time and knee laxity analysis. The comprehensive methodological approach for quantifying the relationship between the status of the soft tissues within the knee joint and its biomechanical condition is quite important in assessing both the trauma and the reconstruction effects. However, the relationship between cartilage T2 values and knee laxity are not reliable. The conclusion is not clear, and the represented statistical analysis ways are weird. This manuscript is very difficult to understand and not summarized well.  Therefore it is not suitable for publication. The comments are below.

  1. Check line 89, "2. Methods"
  2. Fig.1 and Fig2, no sample information.
  3. Fig.3, "&"no statistical information. The results were not explained well.
  4. In Table 2 and Table 3, statistical difference symbols are confused.
  5. 3.3 and 3.4, readers do not know where come from p and R-value.
  6. Discussion is very confusing and hard to understand.

Reviewer 3 Report

This manuscript by Marchiori et al. entitled “A comprehensive framework to evaluate the effects of anterior cruciate ligament injury and reconstruction on graft and cartilage status through the analysis of MRI T2 relaxation time and knee laxity” investigates the association between joint functions and the surrounding tissues of the knee with regards to ACL reconstruction pre- and post-operatively. The authors tried to construct a framework in order to assess the effects of ACL trauma and evaluate the efficacy of surgery using a comprehensive approach. The topic is very interesting and the manuscript is well written. The conclusions are well supported by the results and are well discussed. Some points must be addresses:

  1. In the title a word/phrase such as preliminary or pilot etc should be included to reflect the nature of the study since the major limitation is the sample size.
  2. How the authors assessed skeletal maturity since it is included in the exclusion criteria? Was chronological age a factor that affect the study?
  3. How gender affected the results? Comments for this should be also included in the discussion.
  4. The presentation of the results can be improved by merging (eg. Fig3 and Table 2, Tables 3 and 4 where appropriate). In addition, the most interesting results from sections 3.3-3.5 should be graphically presented and not only described.
  5. The discussion is explicit but could be written more concisely.
  6. The text needs a moderate editing for spelling and grammar errors.

Round 2

Reviewer 1 Report

Thank you for all answers and corrections. I believe it improved a paper. 

Author Response

Thanks again for the comments, which helped us to improve the manuscript.

Reviewer 2 Report

The authors addressed all my previous concerns. 

Author Response

We thank the reviewer for those comments, which helped us to improve the manuscript.

The English language has been deeply checked.